# PDEBench: An Extensive Benchmark for Scientific Machine Learning

**Makoto Takamoto***
NEC Labs Europe

**Timothy Praditia**†
University of Stuttgart

**Raphael Leiteritz**
University of Stuttgart

**Dan MacKinlay**
CSIRO's Data61

**Francesco Alesiani**
NEC Labs Europe

**Dirk Pflüger**
University of Stuttgart

**Mathias Niepert**
University of Stuttgart

## Abstract

Machine learning-based modeling of physical systems has experienced increased interest in recent years. Despite some impressive progress, there is still a lack of benchmarks for Scientific ML that are easy to use but still challenging and representative of a wide range of problems. We introduce PDEBench, a benchmark suite of time-dependent simulation tasks based on Partial Differential Equations (PDEs). PDEBench comprises both code and data to benchmark the performance of novel machine learning models against both classical numerical simulations and machine learning baselines. Our proposed set of benchmark problems contribute the following unique features: (1) A much wider range of PDEs compared to existing benchmarks, ranging from relatively common examples to more realistic and difficult problems; (2) much larger ready-to-use datasets compared to prior work, comprising multiple simulation runs across a larger number of initial and boundary conditions and PDE parameters; (3) more extensible source codes with user-friendly APIs for data generation and baseline results with popular machine learning models (FNO, U-Net, PINN, Gradient-Based Inverse Method). PDEBench allows researchers to extend the benchmark freely for their own purposes using a standardized API and to compare the performance of new models to existing baseline methods. We also propose new evaluation metrics with the aim to provide a more holistic understanding of learning methods in the context of Scientific ML. With those metrics we identify tasks which are challenging for recent ML methods and propose these tasks as future challenges for the community. The code is available at https://github.com/pdebench/PDEBench.

## 1 Motivation

In the emergent area of *Scientific Machine Learning* (or *machine learning for physical sciences* or *data-driven science*), recent progress has broadened the scope of traditional machine learning (ML) methods to include the time-evolution of physical systems. Within this field, rapid progress has been made in the use of neural networks to make predictions using functional observations over continuous domains [8, 46] or with challenging constraints and with physically-motivated conservation laws [34, 61, 47]. These neural networks provide an approach to solving PDEs complementing traditional

---

*E-mail:Makoto.Takamoto@neclab.eu     †E-mail:timothy.praditia@iws.uni-stuttgart.de

36th Conference on Neural Information Processing Systems (NeurIPS 2022) Track on Datasets and Benchmarks.

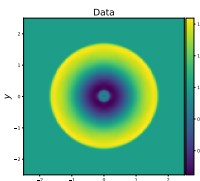 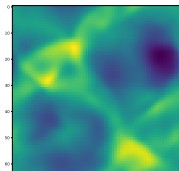 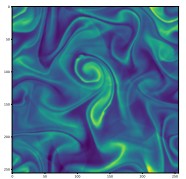 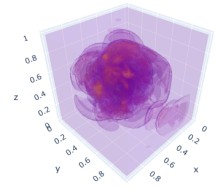

Figure 1: PDEBENCH provides multiple non-trivial challenges from the Sciences to benchmark current and future ML methods, including wave propagation and turbulent flow in 2D and 3D

numerical solvers. For instance, data-driven ML methods are useful when observations are noisy or the underlying physical model is not fully known or defined [11]. Moreover, neural models have the advantage of being continuously differentiable in their inputs, a useful property in several applications. In physical system design [1], for instance, the models are themselves physical objects and thus not analytically differentiable. Similarly, in many fields such as hydrology [14], benchmark physical simulation models exist but the forward simulation models non-differentiable black boxes. This complicates optimisation, control, sensitivity analysis, and solving inverse inference problems. While complex methods such as Bayesian optimisation [38, 50, 42] or reduced order modelling [16] are in part an attempt to circumvent this lack of differentiability, gradients for neural networks are readily available and efficient.

For classical ML applications such as image classification, time series prediction or text mining, various popular benchmarks exist, and evaluations using these benchmarks provides a standardised means of testing the effectiveness and efficiency of ML models. As yet, a widely accessible, practically simple, and statistically challenging benchmark with ready-to-use datasets to compare methods in Scientific ML is missing. While some progress towards reference benchmarks has been made in recent years (see section 2), we aim to provide a benchmark that is more comprehensive with respect to the PDEs covered and which enables more diverse methods for evaluating the efficiency and accuracy of the ML method. The problems span a range of governing equations as well as different assumptions and conditions; see Figure 1 for a visual teaser. Data may be generated by executing code through a common interface, or by downloading high-fidelity datasets of simulations. All code is released under a permissive open source license, facilitating re-use and extension. We also propose an API to ease the implementation and evaluation of new methods, provide recent competitive baseline methods such as FNOs and autoregressive models based on the U-Net, and a set of pre-computed performance metrics for these algorithms. We may thus compare their predictions against the "ground truth" provided by baseline simulators used to generate the data.

As in other machine learning application domains, benchmarks in Scientific ML may serve as a source of readily-available training data for algorithm development and testing without the overhead of generating data *de novo*. In these emulation tasks, the training/test data is notionally unlimited, since more data may always be generated by running a simulator. However, in practice, producing such datasets can have an extremely high burden in compute time, storage and in access to the specialised skills needed to produce them. PDEBENCH also addresses the need for quick, off-the-shelf training data, bypassing these barriers while providing an easy on-ramp to be extended.

In this work, we propose a versatile benchmark suite for Scientific ML (a) providing diverse data sets with distinct properties based on 11 well-known time-dependent and time-independent PDEs, (b) covering both "classical" forward learning problems and inverse settings, (c) all accessible via a uniform interface to read/store data across several applications, (d) extensible, (e) with results for popular state-of-the-art ML models (FNO, U-Net, PINN) for (f) a set of metrics that are better-suited for Scientific ML, (g) with both data to download and code to generate more data, and (h) pre-trained models to compare against. The inverse problem scenarios comprise initial and boundary conditions and PDE parameters (e.g. viscosity). Each data set has a sufficiently large number of samples for training and test, for a variety of parameter values, with a resolution high enough to capture local dynamics. As an additional note, our goal is not to provide a complete benchmark that includes all possible combinations of inference tasks on all known experiments, but rather to ease the task for subsequent researchers to benchmark their favoured methods. Part of our goal here is to invite other researchers to fill in the gaps for themselves by leveraging our ready-to-run models.

To evaluate ML methods for scientific problems, we consider several metrics that go beyond the standard RMSE and include properties of the underlying physics. The initial experimental results we obtained using PDEBENCH confirm the need for comprehensive Scientific ML benchmarks: There is no one-size-fits-all model and there is plenty of room for new ML developments. The results show that the standard error measure in ML, the RMSE on test data, is not a good proxy for evaluating ML models, in particular in turbulent and non-smooth regimes where it fails to capture small spatial scale changes. We furthermore cover an application where a parameter of the underlying PDE heavily influences the difficulty of the problem for ML baselines. We also observe unexpected experimental results for the generalization behavior of auto-regressive training, which seems to be more challenging in the scientific realm. In the remainder of the paper, we first address related work, introduce PDEBENCH and the underlying design choices, and discuss results for a few selected experiments.

## 2  Related Work

PDE benchmarking has particular challenges. Unlike many classic datasets, PDE datasets can be large on a gigabyte or terabyte scale and still contain only few data points. And unlike monolithic benchmark datasets such as ImageNet, the datasets for each PDE approximation task are specific to that task. Each set of governing equations or experiment design assumptions leads to a distinct dataset of PDE samples. Recent works in PDEs have attempted to produce standardised datasets covering well-known challenges [44, 19, 51]. [19] targets non-ML uses. [51] is specialised for particular classes of equations. Of these, the excellent work of [44] is most closely related, but with only four physical systems, it still lacks sufficient scale and diversity of data to challenge emerging ML algorithms. We expand the range of benchmarks in this domain by providing a larger, more diverse problem selection and scale than these previous attempts (11 PDEs with different parametrizations leading to 35 datasets). We additionally consider inverse problems for PDEs [53, 56], with the goal to identify unobserved latent parameters using ML. This has not been covered by benchmarks so far, despite its increasing importance in the community. Furthermore, most work in this scope considers classical statistical error measures such as the RMSE over the whole domain and at most PDE-motivated variants such as the RMSE of the gradient [44]. Measures based on properties of the underlying physical systems, as studied in this work, are lacking.

An overview and taxonomy of Scientific ML developments can be found in [28, 6]. For developing our baselines, we focus on using neural network models to approximate the outputs of some *ground truth* PDE solver, given data generated by that solver, which itself aims to directly implement the numerical solution of a given partial differential equation. A range of methods aim to solve problems fitting this description, reviewed in [22]. Methods include Physics-informed neural networks (PINNs) [47], Neural operators (NOs) [32, 26], treating ResNet as a PDE approximant [49], custom architectures for specific problems such as TFNet for turbulent fluid flows [61], and generic image-to-image regression models such as the U-Net [48]. These approaches each have different assumptions, domains of applicability, and data processing requirements. For a more comprehensive discussion of prior work in Scientific ML we refer the reader to Appendix A.

## 3  PDEBENCH: A Benchmark for Scientific Machine Learning

In the following we describe the general learning problem addressed with the benchmark, the currently covered PDEs, existing implemented baselines (all developed using PyTorch [45], and PINN specifically using DeepXDE [34]), and the ways in which the benchmark follows FAIR data principles [63].

### 3.1  General Problem Definition

A solution to a PDE is a vector-valued function $\mathbf{v} : \mathcal{T} \times \mathcal{S} \times \Theta \to \mathbb{R}^d$ on some spatial domain $\mathcal{S}$, with temporal index $\mathcal{T}$, and some possibly function-valued parameter-space $\Theta$. For example in a heat diffusion equation, $\mathbf{v}$ might represent the local temperature $\tau \in \mathbb{R}^1$ of some substrate at some given point $\mathbf{s} \in \mathcal{S}$, at a given moment $t \in \mathcal{T}$, and conditional upon a spatially-varying scalar conductivity field representing an inhomogeneous substrate $\theta : \mathcal{S} \to \mathbb{R}^+$. The operator mapping from the state of

Table 1: Summary of PDEBENCH's datasets with their respective number of spatial dimensions $N_d$, time dependency, spatial resolution $N_s$, temporal resolution $N_t$, and number of samples generated.

| PDE | $N_d$ | Time | $N_s$ | $N_t$ | Number of samples |
|---|---|---|---|---|---|
| advection | 1 | yes | 1 024 | 200 | 10 000 |
| Burgers' | 1 | yes | 1 024 | 200 | 10 000 |
| diffusion-reaction | 1 | yes | 1 024 | 200 | 10 000 |
| diffusion-reaction | 2 | yes | $128 \times 128$ | 100 | 1000 |
| diffusion-sorption | 1 | yes | 1 024 | 100 | 10 000 |
| compressible Navier-Stokes | 1 | yes | 1 024 | 100 | 10 000 |
| compressible Navier-Stokes | 2 | yes | $512 \times 512$ | 21 | 1000 |
| compressible Navier-Stokes | 3 | yes | $128 \times 128 \times 128$ | 21 | 100 |
| incompressible Navier-Stokes | 2 | yes | $256 \times 256$ | 1000 | 1000 |
| Darcy flow | 2 | no | $128 \times 128$ | – | 10 000 |
| shallow-water | 2 | yes | $128 \times 128$ | 100 | 1000 |

the solution at one timestep to the solution one time step later, $\mathfrak{F}_\theta : \mathbf{v}_\theta(t, \cdot) \to \mathbf{v}_\theta(t + 1, \cdot)$ is referred to as the *forward propagator*.

The objective of Scientific ML is to find some ML-based surrogate, sometimes referred to as an emulator, of this forward propagator by learning an approximation $\widehat{\mathfrak{F}}_\theta \simeq \mathfrak{F}_\theta$. The forward propagator of a PDE is dependent not only on the current state, but also upon both spatial and temporal derivatives of the state field. In practice, temporal derivatives of solutions are often not conveniently encoded by system states at one single time step. Hence, the forward propagator may also depend on multiple previous timesteps of the solution, enabling finite-difference approximations of the temporal derivatives. The discretised forward propagator $\mathring{\mathfrak{F}}_\theta$ then operates on $\ell \geq 1$ consecutive timesteps so that $\mathring{\mathfrak{F}}_\theta : \mathbf{v}_\theta(t - \ell, \cdot), \ldots, \mathbf{v}_\theta(t - 1, \cdot) \mapsto \mathbf{v}_\theta(t, \cdot)$, which is abbreviated as $\mathbf{v}_\theta([t-\ell{:}t-1], \cdot) := \mathbf{v}_\theta(t - \ell, \cdot), \ldots, \mathbf{v}_\theta(t - 1, \cdot)$.

We seek to approximate this discretised operator with an emulator $\widehat{\mathring{\mathfrak{F}}}_\theta \simeq \mathring{\mathfrak{F}}_\theta$ in the sense that predictions the emulator makes should be close to the ground truth simulation given the same inputs, with respect to some measure of cost. We fix a parametric class of models $\{\mathfrak{F}_{\theta,\phi}\}_\phi$. From this class we *learn* a surrogate $\widehat{\mathring{\mathfrak{F}}}_{\theta,\phi}$ from data. In learning, we take a dataset $\mathcal{D}$ comprising discretized PDE solutions conditional on selected parameter values $(\theta_k)$, $\mathcal{D} := \{\mathbf{v}_{\theta_k}^{(k)}([0{:}t_{\max}], \cdot) \mid k = 1, \ldots, K\}$. Fixing a loss functional $L$, we aim to find some $\phi$ achieving a minimal total loss on the training dataset

$$\widehat{\phi} = \mathrm{argmin}_\phi \sum_{t=1}^{t_{\max}} \sum_{k=1}^{K} L\left( \mathfrak{F}_{\theta_k,\phi}\{\mathbf{v}_{\theta_k}^{(k)}([t-\ell{:}t-1], \cdot)\}, \mathbf{v}_{\theta_k}^{(k)}(t, \cdot) \right). \tag{1}$$

Due to the use of iterative optimization algorithms such as stochastic gradient descent and the non-convex nature of the above optimization problem, we typically obtain local optima. $\mathcal{D}$ is generated by a ground-truth solver designed to simulate the desired dynamics with high precision. In this data we may vary initial conditions, that is, varying $\mathbf{v}_\theta(0, \cdot)$, varying $\theta$, or both.

In addition to the forward problem, we also consider the use of learned surrogate models to approximately solve *inverse problems* [53, 56], where an unknown initial condition $\mathbf{v}_\theta(0, \cdot)$ or unknown parameter $\theta$ is chosen to be congruent with some observed outputs $\mathbf{v}_\theta([t{:}t + \ell], \cdot)$. We follow [32, 35] in using an approximate surrogate approach, taking the forward surrogates as mean predictors for the model. We assume $\mathbf{v}_\theta(t, \cdot) = \mathfrak{F}_{\theta,\phi}\{\mathbf{v}_\theta([t-\ell{:}t-1], \cdot)\} + \epsilon$ for some mean-zero observation noise $\epsilon$, and assuming a prior distribution for the unknown of interest. Other inversion methods can be used in this domain, such as generative adversarial models, [9] or variational autoencoders [54].

## 3.2 Overview of Datasets and PDEs

The current version of the benchmark provides datasets generated for various PDEs ranging from 1 to 3 dimensional spatial domains. There are both time-dependent and time-independent PDEs. The current datasets are summarized in Table 1; see Figure 1 for a visual teaser. Each sample is generated with different parameters, initial conditions, and boundary conditions. Generalization to different

parameters, varying initial conditions, and proper treatment of complex boundary conditions are still open challenges in Scientific ML [21, 4, 2, 29]. The parameters which are varied to provide several datasets include the advection speed in the advection equation, the forcing term in the Darcy flow, as well as the viscosity in the Burgers' and compressible Navier-Stokes equations all of which can lead to significantly different behaviors of the simulated systems. Additionally, besides the periodic boundary condition that is most commonly used in Scientific ML studies, we also provide datasets generated with the Neumann boundary condition in the 2D diffusion-reaction and shallow-water equations, the Cauchy boundary condition in the diffusion-sorption equation, and the Dirichlet condition in the incompressible Navier-Stokes equation.

We designed this benchmark to represent a diverse set of challenges for emulation algorithms. In particular, we focus on hydromechanical field equations. Following this philosophy, we selected 6 basic and 3 advanced real-world problems. The basic PDEs are stylized, simple models: 1D advection/Burgers/Diffusion-Reaction/Diffusion-Sorption equations, 2D Diffusion-Reaction equation, and 2D DarcyFlow; the advanced and real-world PDEs incorporate features of real-world modeling tasks: Compressible and incompressible Navier-Stokes equations, and shallow-water equations. The PDEs exhibit a variety of behaviors of real-world significance which are known to challenge emulators, such as sharp shock formation dynamics, sensitive dependence on initial conditions, diverse boundary conditions, and spatial heterogeneity. Finding a surrogate model which can approximate these challenging dynamics with high fidelity we argue is a necessary precondition to applying such models in the real world. While some of these have been used in prior work, a publicly available benchmark dataset is, to the best of our knowledge, not available.

In the following we provide a brief introduction and important features of the advanced PDEs. More detailed explanations for all the PDEs are provided in Appendix D.

**Compressible Navier-Stokes equations**    The compressible fluid dynamics equations describe a fluid flow whose expression is given as:

$$\partial_t \rho + \nabla \cdot (\rho \mathbf{v}) = 0, \quad \rho(\partial_t \mathbf{v} + \mathbf{v} \cdot \nabla \mathbf{v}) = -\nabla p + \eta \triangle \mathbf{v} + (\zeta + \eta/3)\nabla(\nabla \cdot \mathbf{v}), \quad \text{(2a)}$$

$$\partial_t(\epsilon + \rho v^2/2) + \nabla \cdot [(p + \epsilon + \rho v^2/2)\mathbf{v} - \mathbf{v} \cdot \sigma'] = \mathbf{0}, \quad \text{(2b)}$$

where $\rho$ is the mass density, $\mathbf{v}$ is the fluid velocity, $p$ is the gas pressure, $\epsilon$ is an internal energy described by the equation of state, $\sigma'$ is the viscous stress tensor, and $\eta$ and $\zeta$ are shear and bulk viscosity, respectively. This equation can describe more complex phenomena, such as shock wave formation and propagation. It is applied to many real-world problems, such as the aerodynamics around airplane wings and interstellar gas dynamics.

**Incompressible Navier-Stokes equations**    The Navier-Stokes equation is the incompressible version of the compressible fluid dynamics equation, applicable to sub-sonic flows. This equation can model a variety of systems, from hydromechanical systems to weather forecasting or investigating turbulent dynamics.

**Shallow-Water Equations**    The shallow-water equations, derived from the compressible Navier-Stokes equations, present a suitable framework for modeling free-surface flow problems. In 2D, these come in the form of the following system of hyperbolic PDEs,

$$\partial_t h + \nabla h \mathbf{u} = 0, \quad \partial_t h \mathbf{u} + \nabla \left( \mathbf{u}^2 h + \frac{1}{2} g_r h^2 \right) = -g_r h \nabla b, \quad \text{(3a)}$$

where $\mathbf{u} = u, v$ being the velocities in the horizontal and vertical direction, $h$ describing the water depth, and $b$ describing a spatially varying bathymetry. $h\mathbf{u}$ can be interpreted as the directional momentum components and $g_r$ describes the gravitational acceleration. The mass and momentum conservation properties even hold across shocks in the solution and thus challenging datasets can be generated. Example applications include the simulation of tsunamis or general flooding events.

### 3.3   Overview of Metrics

The standard approach of computing the RMSE on test data falls short of capturing important optimization criteria in Scientific ML. A good fit to (often sparse) data is not sufficient if the physics of the underlying problem is severely violated. Physics-informed learning that aims to conserve

```
from pyDaRUS import Dataset
p_id = "doi:10.18419/darus-2986"
dataset = Dataset.from_dataverse_doi(p_id, filedir="data/")
```

Listing 1: Including a benchmark dataset.

physical quantities must therefore be evaluated with appropriate metrics. A global, averaged metric for instance cannot capture small spatial scale changes critical in turbulent regimes. Moreover, a single evaluation metric is not sufficient to compare different methods with respect to their ability to extrapolate to unseen time steps and parameters which are important but underexplored evaluation criteria for ML surrogates. Hence, the proposed benchmark includes several novel metrics which we believe provide a deeper and more holistic understanding of the surrogate's behavior and which are designed to reflect both the data and physics perspective. The following table summarizes the metrics used; further details can be found in the Appendix B.

| Scope | Acronym | Metric |
|-------|---------|--------|
| Data view | RMSE | root-mean-squared-error |
| | nRMSE | normalized RMSE (ensuring scale independence) |
| | max error | maximum error (local worst case; also proxy for stability of time-stepping) |
| Physics view | cRMSE | RMSE of conserved value (deviation from conserved physical quantity) |
| | bRMSE | RMSE on boundary (whether boundary condition can be learned) |
| | fRMSE low | RMSE in Fourier space, low frequency regime (wavelength dependence) |
| | fRMSE mid | RMSE in Fourier space, medium frequency regime |
| | fRMSE high | RMSE in Fourier space, high frequency regime |

### 3.4 Existing Baseline Surrogate Models

**U-Net**  U-Net [48] is an auto-encoding neural network architecture used for processing images using multi-resolution convolutional networks with skip layers. U-Net is a black-box machine learning model that propagates information efficiently at different scales. Here, we extended the original implementation, which uses 2D-CNN, to the spatial dimension of the PDEs (i.e. 1D,3D).

**Fourier neural operator (FNO)**  FNO [32] belongs to the family of Neural Operators (NOs), designed to approximate the forward propagator of PDEs. FNO learns a resolution-invariant NO by working in the Fourier space and has shown success in learning challenging PDEs.

**Physics-Informed Neural Networks (PINNs)**  Physics-informed neural networks [47] are methods for solving differential equations using a neural network $u_\theta(t, x)$ to approximate the solution by turning it into a multi-objective optimization problem. The neural network is trained to minimize the PDE residual as well as the error with regard to the boundary and initial conditions. PINNs naturally integrate observational data [30], but require retraining for each new condition.

**Gradient-Based Inverse Method**  Since the surrogate model is fully differentiable, we use its gradient to solve inverse inference by minimizing the prediction loss [7, 40], where a function surface defining the unknown initial conditions [35], is specified through bilinear interpolation.

### 3.5 Data Format, Benchmark Access, Maintenance, and Extensibility

The benchmark consists of different data files, one for each equation, type of initial condition, and PDE parameter, using the HDF5 [15] binary data format. Each such file contains multiple arrays where each array has the dimensions $N, T, X, Y, Z, V$ with $N$ the number of samples, $T$ the number of time steps, and $X, Y, Z$ the spatial dimensions and $V$ the dimension of the field. Additional information on the data format is provided in the Supplementary Material.

PDEBENCH's datasets are stored and maintained using DARUS, the University of Stuttgart's data repository based on the OpenSource Software DataVerse[3]. DARUS follows the Findable, Accessible, Interoperable and Reusable (FAIR) data principles [63]. All data uploaded to DaRUS gets a DOI as

---

[3] https://dataverse.org

```
from pdebench.fno.utils import FNODatasetSingle
filename = "data/2D_diff-react_NA_NA"
train_data = FNODatasetMult(filename)
train_loader = torch.utils.data.DataLoader(train_data)
```

Listing 2: Using the PyTorch data loader.

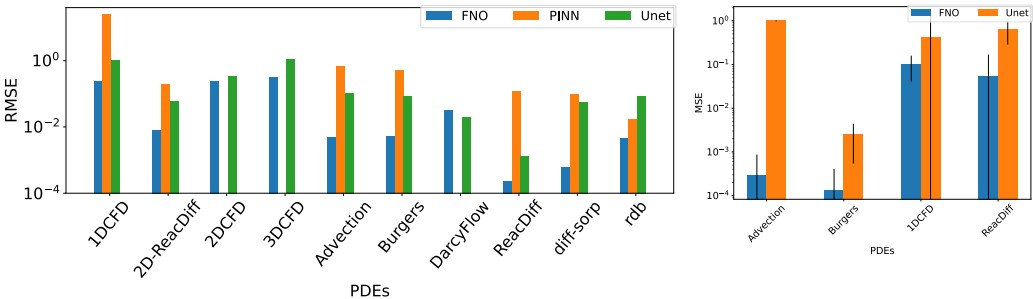

Figure 2: Comparisons of baseline models' performance for different problems for (a) the forward problem and (b) the inverse problem.

a persistent identifier, a license, and can be described with an extensive set of metadata, organized in metadata blocks. A dedicated team ensures that DARUS is continuously maintained. Through DARUS we provide a permanent DOI (doi:10.18419/darus-2986) [55] for the benchmark data. We also support a straightforward inclusion of the benchmark with a few lines of code. In Listing 1 we demonstrate the way in which the Dataverse [10] platform⁴ supports the integration of pre-generated datasets using a few lines of code. Specifically, we utilize the easy-to-use pyDaRUS Python package to access the data. It provides a simple API for both downloading and uploading data as well as providing metadata to the Dataverse platform.

In Listing 2 we show an example leveraging pre-defined classes included in our benchmark code to load specific datasets as PyTorch [45] `Dataset` classes. Subsequently, these can be used to construct common `DataLoader` instances for training custom ML models. We utilize the Hydra [64] library simplifying the configuration of both surrogate model training as well as the generation of additional datasets. For the latter, we provide and expose various parameters of the underlying simulations for the end user to tweak. This provides a low barrier of entry for users to try out benchmarking with new experiments or baseline configurations.

# 4 A Selection of Experiments

In this section, we present a selection of experiments for the PDEBENCH datasets. An exhaustive discussion of all results is beyond the scope of this paper. An extensive set of additional results, tables, and plots can be found in the Appendix.

## 4.1 Baseline Setups

We trained and tested the baseline emulator models, namely U-Net, FNO, and PINN with the datasets generated with the PDEs described in subsection 3.2. The data was split into training and test data: 90% was for training and 10% for test data. For FNO, we followed the original implementation, hyperparameters, and training protocols. We trained U-Net similarly to FNO, but with the autoregressive methods with the pushforward trick with slight modification to the original implementation [4]. A more comprehensive comparison between different U-Net training methods is presented in subsection 4.3. The PINN baseline is implemented using the open-source DeepXDE [34] library. The training was performed on GeForce RTX 2080 GPUs for 1D/2D cases, and GeForce GTX 3090 for 3D cases. The detailed training protocol and hyper-parameters are provided in Appendix C.

---

⁴ https://darus.uni-stuttgart.de/dataverse/sciml_benchmark

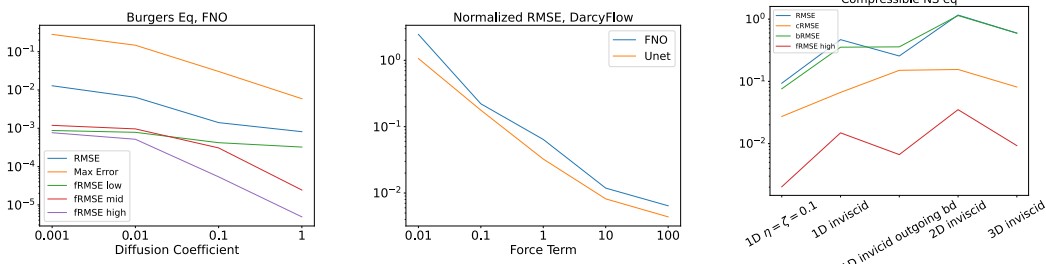

Figure 3: Detailed visualization of (a) Burgers', (b) DarcyFlow, and (c) Compressible NS eqs.

Training code and configurations are open and well documented, allowing researchers to easily reproduce or extend these methods.

## 4.2 Baseline Performance

Figure 2 [5] visualizes the RMSE performance of the surrogate models, averaged for each trained model over different PDE parameters. A more detailed comparison is shown in Appendix E. [6] Among the baseline surrogate models, FNO provides the best prediction for most metrics. It learns the differential operators well, leading to low errors even for the conserved quantities and on the boundaries. Additionally, FNO has a consistent error of about $4 \times 10^{-4}$ across the frequency spectrum for many problems highlighting its ability to learn in Fourier space. As an example, Figure 4 depicts the FNO and U-Net predictions for the 2D diffusion-reaction equation and the training data obtained from a numerical simulator. Our baseline results further indicate that the PINNs might deal better than expected with high-frequency features, despite prior observations [62]. As an example for an inverse problem setup, we identify the initial condition to minimize the prediction error of the ML surrogate over a 15 time steps horizon. In Figure 2b, we present the MSE of the prediction of estimated initial condition (with error bars) for 4 of the 11 datasets (1D). The results show that FNO outperforms U-Net also for the inverse problem. However, our benchmark also reveals several tasks which these methods cannot treat properly. First, Figure 3a shows that the FNO's error increases with decreasing diffusion coefficient where a strong discontinuity appears. This can be attributed to Gibb's phenomenon for FNO's limited maximum wave frequency in Fourier space, as shown by an increase of two orders of magnitude for high-frequency fRMSE. Second, Figure 3b shows that the normalized RMSE increases with decreasing force term, which is equivalent to decreasing the scale-value of the solution (in our case, force term 0.01 means $\text{mean}(|u|) \approx 0.01$).[7] Third, Figure 3c shows several metrics for the compressible Navier-Stokes equations. It shows the overall RMSE is very bad in comparison to the basic PDEs, such as the Burgers equation. Interestingly, the 3D inviscid case shows lower error than 2D inviscid case. We posit this is due to lower resolution resulting in smooth train/validation samples which FNO can learn very efficiently. This also indicates that high-resolution training samples should be used to create a surrogate model for real-world problems with a Reynolds number of more than $10^6$.

## 4.3 Temporal Error Analysis

**Autoregressive Behaviour of U-Net** We observed instabilities when training U-Net with fully autoregressive mode. When trained in an open loop, i.e. only 1 time step ahead without feedback of prediction, the error during testing quickly accumulates with more unrolled time steps. Therefore, we tried three different training methods as described in the previous section. Figure 5a shows the RMSE behaviour calculated at different unrolled time steps. It shows that for different U-Net training strategies, the RMSE behaviors are different. We observed that U-Net with the pushforward trick

---

[5] In Figure 2 CFD means compressible fluid dynamics or, equivalently, the compressible Navier-Stokes equations.
    [6] Note that we omitted the PINN baseline score for the 2D/3D Compressible Navier-Stokes equations and the DarcyFlow. The reason for the former case is limited GPU memory, and the reason for the latter is that the DarcyFlow's problem setup is to learn the mapping from diffusion coefficients $a(x)$ to a steady state solution, which cannot be treated by PINNs.     [7] Note that U-Net is consistently better than FNO in this case. We postulate that this is due to the strong similarity between this task and the diffusion-like regression task that the original U-Net targets.

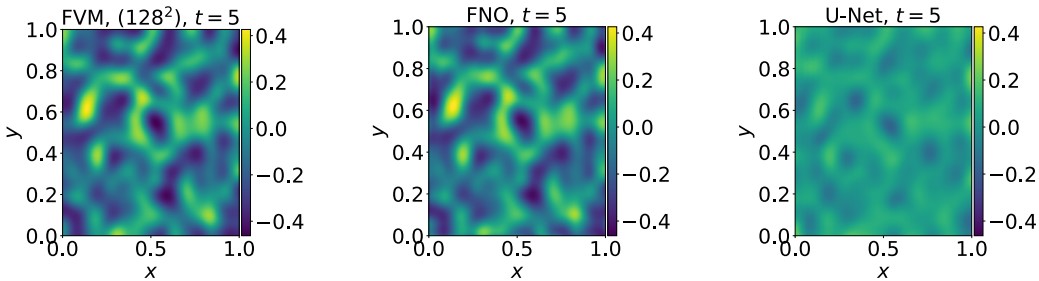

Figure 4: (a) Visualization of the 2D diffusion-reaction data generated with a standard finite volume (FVM) solver and a resolution of $128^2$, (b) FNO prediction, and (c) U-Net prediction.

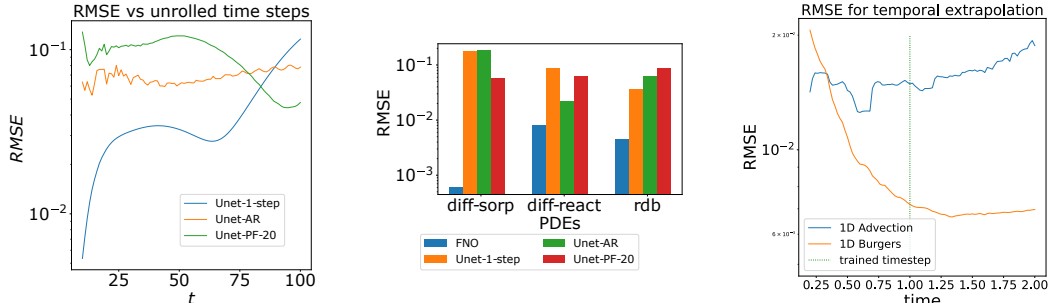

Figure 5: (a) Plots of the RMSE calculated at different unrolled time steps, (b) comparison of each autoregressive method, and (c) RMSE for temporal extrapolation.

provides better stability during training for all problems, and better accuracy for a longer prediction horizon. For this reason, we used the U-Net with pushforward trick as our baseline score for U-Net.

**Temporal Extrapolation** Figure 5c plots the RMSE temporal evolution of 1D Advection and Burgers equations predicted using the FNO model. Different from the other scenarios, we trained only until half of the time steps used in the other cases (the green dotted line in the figure). The main purpose of this experiment is to test how well the ML model (FNO) learns the temporal dependency of the PDEs with limited information. We observe monotonically increasing errors after the time steps used in training ($t > 1$). This indicates that the present ML models cannot properly capture the dynamic behavior of the PDEs, and it is a challenging task to provide reliable predictions beyond the time experienced during training.

### 4.4 Inference Time Comparison

In this subsection, we provide runtime comparisons between the numerical PDE solvers that we use to generate a single data sample and the FNO, the most accurate ML model according to our experiments. A fair runtime comparison between classical solvers and ML methods is challenging because these method classes are usually optimized for different hardware setups. In this paper, we provide the information on the hardware and system configuration in Appendix F [8]. In addition, we list the ML models' total training time. All the runtime numbers are given in seconds. The highest computational demand originates from the ML training time. However, once the ML models were trained, the ML models' predictions can be computed multiple orders of magnitudes more efficiently. A more complete overview of these experiments can be found in Appendix F. The ML model allows us to predict a solution even in the strong-viscous regime ($\eta = \zeta = 0.1$) efficiently since it eliminates the stability restriction defined by the Courant-Friedrichs-Lewy condition [31] [9]. A more detailed analysis of the resolution sensitivity of inference time is provided in Appendix G.

---

[8] We used the same hardware resources and system configuration for the 2D/3D CFD simulations and the ML methods. Whether or not this consitutes a fair comparison is surely debatable. [9] Note that in the table the ML inference and training time does not monotonically increase with spatial-dimension. This is because the 2D/3D cases' time-step numbers and training sample numbers is much smaller than Diffusion-sorption case.

| PDE | Resolution | Simulation time | ML inference time | ML training time |
|---|---|---|---|---|
| Diffusion-sorption | $1\,024^1$ | 59.83 | 0.32 | 48 760 |
| 2D CFD ($\eta = \zeta = 0.1$) | $512^2$ | 582.61 | 0.14 | 107 567 |
| 3D CFD (inviscid) | $64^3$ | 60.06 | 0.14 | 12 387 |

## 5  Conclusions and Limitations

With PDEBENCH we contribute an extensive benchmark suite for comparing and evaluating methods on the realm of Scientific ML. We provide both pre-computed datasets for easy access in a dataverse as well as code to generate new data from configurable simulation runs. The focus is on time-dependent PDE problems ranging from simple 1D equations to challenging 3D coupled systems of equations featuring challenging boundary conditions. Furthermore, we present a variety of different evaluation metrics in order to better understand strengths and weaknesses of the machine learning methods in a scientific computing context. We also provide an example application for utilizing Scientific ML methods for inverse modeling with our benchmark data. We believe this will be an important area in the future for machine learning models to produce competitive results both in accuracy as well as runtime when compared to numerical methods.

**Limitations**   While PDEBENCH provides an easy-to-use, modular and extensible approach to devising and testing ML surrogates for PDE simulations, the scope of our benchmark is naturally limited. Our main focus is on time-dependent PDEs for different types of flow problems which pose a wide variety of challenges to Scientific ML. We currently do not cover other types of physics nor quantum mechanics as this goes beyond the scope of this paper. With respect to flow problems, our main limitations are that we do not yet cover multi-phase flows or non-rectangular domains. This is left for future work.

## Acknowledgements

We thank MinMae (John) Kim, Ran Zhang, Tianqi Wang, Yizhou Yang, Gefei Shan and Simon Brown of the ANU Techlauncher for consulting on code design and implementation. Partially funded by Deutsche Forschungsgemeinschaft (DFG, German Research Foundation) under Germany's Excellence Strategy - EXC 2075 – 390740016. We acknowledge support by the Stuttgart Center for Simulation Science (SimTech). We further thank the DaRUS-team, in particular Jan Range.

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
