# OpenReview forum: "PDEBench: An Extensive Benchmark for Scientific Machine Learning"
_NeurIPS.cc/2022/Track/Datasets_and_Benchmarks — NeurIPS 2022 Datasets and Benchmarks _

### Official Review · Reviewer_dRaf · 2022-07-10
**Well-needed benchmark with some room to improve**

**Rating:** 7
**Confidence:** 3
**Correctness:** Yes
**Clarity:** Yes

**Strengths:**

+ This paper is well-positioned to be a standard in this new field of
  learning the solutions to PDEs.
  It is important to have established families
  of PDEs to have a standardized way of comparing methods.
  Otherwise, many new papers jointly propose new classes of problems
  along with new methods and it is difficult to understand
  the main contribution.
+ The baseline methods considered in this paper
  (U-Net, FNE, PINNs, and a gradient-based inverse method)
  are reasonable and seemingly perform well on all of the PDEs
+ Ablations such as in Figure 3 provide nice insights into
  how methods change as the underlying family of PDEs changes.

**Weaknesses:**

While I am optimistic about the general direction of the paper,
I feel there are a few shortcomings that need to be addressed
to make the paper influential. I am very open to reconsidering
my evaluation of the paper with a discussion on these:

+ It is not clear how impactful better-solving the tasks in
  this paper will be and the paper does not convey how poor or
  well the baselines perform on the problems beyond the error metrics.
  I would find it insightful to visualize the solution quality.
+ While the paper focuses on error metrics, I'm also very interested
  in understanding how the computational time compares between methods.
+ I would also find it useful to add as baselines standard ("classical")
  PDE solvers for all of the tasks and report their error metrics
  and runtimes.
+ It's not clear how varied the PDEs are within each family and
  I would again find it useful for intuition to visualize samples
  from the distributions.
+ The paper focuses on the setting where the PDE solution operator
  is regressed onto a ground-truth solution obtained by a high-accuracy
  solver. However there are other interesting classes of methods such
  as the
  [Physics-Informed Neural Operator](https://arxiv.org/pdf/2111.03794.pdf)
  which also work by optimizing the PDE residuals directly that
  would be worthwhile incorporating into a benchmark as well.
+ It's important to connect some of the results back to
  established results published in this space. For example,
  how do the results compare to Figure 3 of the related PDEs in the
  [Fourier Neural Operator](https://arxiv.org/pdf/2010.08895.pdf) paper?
  It would be extremely strong to reproduce their results on
  exactly their settings and then expand beyond that.

**Additional Feedback:**

n/a

**Documentation:**

Yes, and the software practices in the repository look exceptionally well-done

**Relation To Prior Work:**

Yes, see also last comment in weaknesses on connecting the experimental results to related work

**Summary And Contributions:**

This paper proposes a benchmark for learning the solutions to PDEs.
Section 3 describes the basic setup of solution mappings of PDEs
and proposes ~10 classes of PDEs to solve that are representative
of many of the most challenging problems in the community.

---

> ### Author Response · Authors · 2022-08-12
> **Reply to Reviewer #4**
>
> Dear Reviewer #4,
>
> Thank you very much for your kind comment for improving our paper. The following are our responses.
>
> > It is not clear how impactful better-solving the tasks in this paper will be and the paper does not convey how poor or well the baselines perform on the problems beyond the error metrics. I would find it insightful to visualize the solution quality.
>
> Thank you very much for your concerns. We have added several plots of true/FNO/Unet results for all the PDE cases in Figure 4 and the Appendix I. We hope they would help to understand the performance of each model. Note that for the moment we provided the plots only for a subset of PDEs, but we are going to provide for all PDE samples in the camera-ready version.  Additionally, to address your concern about the error metrics, we also use physically-motivated error metrics in this work, which is still missing in previous related works.  Do our changes sufficiently well address your concerns?
>
>
> >While the paper focuses on error metrics, I'm also very interested in understanding how the computational time compares between methods.
>
> We have added a new section discussing inference time of machine learning models and numerical simulation methods in Subsection 4.4 and Appendix F. We also would like to ask you for checking our “General Reply” on this matter.
>
> >I would also find it useful to add as baselines standard ("classical") PDE solvers for all of the tasks and report their error metrics and runtimes.
> It's not clear how varied the PDEs are within each family and I would again find it useful for intuition to visualize samples from the distributions.
>
> Thank you very much for your kind comments on this important matter. We performed a small experiment by generating a higher resolution dataset for the 2D diffusion-reaction problem and downsampled the data to train the ML models. Then, we also generated a lower resolution dataset to be compared to the downsampled data, that we assumed as the ground truth. Detailed information can be found in Appendix H. Additionally, we also added visualizations of some of the PDEs in their corresponding data description in Appendix D. We are welcoming any feedback kindly from you on this matter!
>
>
> >The paper focuses on the setting where the PDE solution operator is regressed onto a ground-truth solution obtained by a high-accuracy solver. However there are other interesting classes of methods such as the Physics-Informed Neural Operator which also work by optimizing the PDE residuals directly that would be worthwhile incorporating into a benchmark as well.
>
> Thank you very much for your kind encouraging comment. Although “Physics-Informed Neural Operator” is one of the most important works in recent years, we feel the method is a little too advanced as a baseline in our paper because one of the motivations of our paper is to provide the basic and important machine learning models even to non-machine learning researchers, such as simulation researchers. Instead, we provided our repository to be able to combine other methods as easily as possible. So we believe interested readers can try advanced methods, e.g., PINO, in our repo. Note that we also cited the PINO paper.
> Do our changes sufficiently well address your concerns, too?
>
>
> >It's important to connect some of the results back to established results published in this space. For example, how do the results compare to Figure 3 of the related PDEs in the Fourier Neural Operator paper? It would be extremely strong to reproduce their results on exactly their settings and then expand beyond that.
>
> Thank you very much for your kind comment. As is pointed out in our paper, we imported the original implementation of FNO, so our FNO model should be able to perfectly reproduce FNO paper’s result. In addition, the current version of the original FNO paper’s dataset generation code is partially implemented in MATLAB, which requires a commercial license, or the python implementation has some compatibility issues. The FNO’s dataset is partially available over google drive, whose availability is linked to the owner account, while we opt for a long-term solution as required by the Dataset call. With respect to the inverse problem, FNO original paper proposes a qualitative study, without providing the accompanying baseline code, while we provide quantitative results and the baseline code.
> We would also like to emphasize that the objective of our paper is to provide a new benchmark dataset for the community, and not to provide a complete benchmark involving all combinations of all possible tasks (please see our general response to all reviewer, regarding problem selection), which we feel would not be possible given the limited page limit and time for rebuttal.
> Again we are welcoming any feedback kindly from you on this matter.

---

> > ### Comment · Reviewer_dRaf · 2022-08-17
> > **Response**
> >
> > (I apologize for my delayed response here, I did not receive an email/notification telling me that you had responded and just saw your comments when manually checking this page.)
> >
> > Thanks for the response and the updated version of the paper, all of these additional details are commendable and exactly what I requested. I have read through all of the reviewing details here and have updated my score from a 4 to a 7 as the response has addressed my concerns. I think the paper is a strong and well-needed contribution for benchmarking PDE solvers and has a high potential to become a standard in the field.
> >
> > One last detail: I still do not have a good sense of what the initial conditions for the PDE instances would look like and would still find it interesting to plot samples from each of them for every domain.

---

> > > ### Author Response · Authors · 2022-08-18
> > > **Reply**
> > >
> > > Thank you very much for your positive response to our rebuttal, and we are really happy that you are satisfied with it now.
> > > For the camera-ready version, we would like to have an additional effort to polish the paper following all the reviewer's encouraging advice, in particular, the ones we cannot fully address because of the time limitation.
> > >
> > > >One last detail: I still do not have a good sense of what the initial conditions for the PDE instances would look like and would still find it interesting to plot samples from each of them for every domain.
> > >
> > > Thank you very much for the suggestion. We are now preparing for the gallery of initial conditions to be put in the Appendix.

---

### Official Review · Reviewer_zkya · 2022-07-23
**Dataset for benchmarking surrogates that approximate PDE**

**Rating:** 7
**Confidence:** 3

**Strengths:**

- Novel dataset for benchmarking Machine Learning model applied to physics.
- Easy to access API.
- Interesting application of the surrogates for solving the inverse problem
- Important ablation for different types of surrogates and metrics to approximate PDE.

**Weaknesses:**

- Diversity: My main concern is about the diversity of the benchmark. There are few datasets (11) and the application domain is mainly hydrodynamics.
- Weak benchmark motivation: In order to motivate further the use of surrogates for scientific ML, and therefore the benchmark, I think it is essential to provide an additional problem setting beside the inverse problem.
- Writing: The authors should consider that they submit to a conference related to Machine Learning, rather than physics. Therefore I think that a nicer introduction to PDE and its importance in modeling physical systems would be great.
- Weak discussion of prior work.



**Additional Feedback:**


I understand that surrogates allow the users to speed up the evaluation of the dynamic systems. Therefore, it is important to have accurate surrogates that model well the underlying system. However, for training the surrogate, you need to spend some time as well, is this not counter-productive? I would like that the authors elaborate more on:
- The advantage of using surrogates. How much is the time gained, for instance: time for simulation vs time for the surrogate training vs time for other solvers, compute resources for simulation vs compute resources for training?
- The sensitivity of the tasks using the surrogates (e.g. solving the inverse problem/setting) to the accuracy of the surrogate. How much additional accuracy do I need to get better solutions for the “downstream” task? Could we have some information about this?


**Clarity:**

The paper is overall clear, but I had to jump back and forth to understand some concepts: e.g. inverse problem (which is defined a couple of pages after its mention).

**Correctness:**

The benchmark construction is clearly explained by presenting the PDE that originated the data. The authors present explicit formulation and motivation for the equations and the associated system.

**Documentation:**

The authors present a Github page with the data and API to access the benchmark.

**Ethics:**

No special concern on this matter.

**Relation To Prior Work:**

The related work is short and could be enriched by listing, for instance, how much data/PDEs/surrogates or which metrics previous work used? LIke that it is easier to understand the position of this work. However, I understand that given the novelty of the settings, related work can be probably scarce.

**Summary And Contributions:**


The authors introduce a dataset to benchmark surrogates that approximate the solution of partial differential equations (PDE). The surrogates can be used for solving related problems, such as the inverse problem, i.e. to find the initial conditions (t=0) of dynamic systems, given information from t>0 about the system. The authors present an ablation of various models using different neural networks, such as U-Net, Frourer Neural Operator, etc. They also study different metrics to evaluate the precision of the prediction of the surrogates. The benchmark is easy to access through an API.

---

> ### Author Response · Authors · 2022-08-12
> **Reply to Reviewer #3**
>
> Dear Reviewer #3,
>
> Thank you very much for your kind comment for improving our paper. The following are our responses.
>
> >Diversity: My main concern is about the diversity of the benchmark. There are few datasets (11) and the application domain is mainly hydrodynamics.
>
> Thank you very much for asking. We would like to emphasize that the number of datasets is not 11, but the number of PDEs is 11. Out of the 11 PDEs, we generated 35 datasets with various parameterizations of the PDEs that lead to different characteristics of the solutions. We now explicitly point this out in the Related Work. To the best of our knowledge, this represents the largest collection of benchmark problems of its kind. In addition, the selected PDEs are representative of a large class of problems, and they cover a large range from very basic ones to advanced real-world problems, such as the Navier-Stokes equations. The range is much larger than provided by previous work. We hope that we provide a critical amount of representative PDEs so that our paper can serve as a nucleus and motivate the community to contribute and extend the range of problems.
> Concerning the domain, we have added a comment in the introduction that we mainly focus on hydrodynamic problems which we believe to be one of the most popular targets for the scientific ML community. We also would like to emphasize that, for example, the 2D-reaction-diffusion equation is not only related to hydrodynamics but modern biological problem formulations and Burgers’ equation is relevant to numerous applications, as discussed in the supplemental material’s Section D.9
>
> >Weak benchmark motivation: In order to motivate further the use of surrogates for scientific ML, and therefore the benchmark, I think it is essential to provide an additional problem setting beside the inverse problem.
>
> Thank you very much for your kind comment. We provide the state-of-the-art implementation of surrogate ML models and in addition provide an implementation for the inverse problem. The aim of our  Benchmark is not to cover all possible problems, but to provide a standard dataset and reference models for the most used scenarios. Our benchmark is highly extensible in both the baseline models and on the generation components, thus being future-proof for new applications. We also would like to ask you for checking our “General Reply” on this matter.
>
> >Writing: The authors should consider that they submit to a conference related to Machine Learning, rather than physics. Therefore I think that a nicer introduction to PDE and its importance in modeling physical systems would be great.
>
> Thank you very much for your kind suggestion. We would like to point out that we have provided a nice introduction of our PDEs in Section 1, Section 3.1, 3.2, as well as more details on the physics models in Appendix D with new references for interested readers.
>
>
> >Weak discussion of prior work. +  The related work is short and could be enriched by listing, for instance, how much data/PDEs/surrogates or which metrics previous work used? LIke that it is easier to understand the position of this work. However, I understand that given the novelty of the settings, related work can be probably scarce.
>
> We have polished our introduction of the PDEs and the “related work” section. We also discussed the metrics used in the previous work compared to our work. Do our changes sufficiently well address your concerns?
>
> >The advantage of using surrogates. How much is the time gained, for instance: time for simulation vs time for the surrogate training vs time for other solvers, compute resources for simulation vs compute resources for training?
>
> Thank you very much for your fruitful comment! We have added a new section discussing inference time comparison between machine learning models and simulation methods in Subsection 4.4 and Appendix F. We also would like to ask you for checking our “General Reply” on this matter.
>
> >The sensitivity of the tasks using the surrogates (e.g. solving the inverse problem/setting) to the accuracy of the surrogate. How much additional accuracy do I need to get better solutions for the “downstream” task? Could we have some information about this?
>
> Thank you very much for your encouraging comment! Although the topic is quite interesting, the limitation of pages and time prohibited us to provide that information. We would like to tackle those topics in our future work.

---

> > ### Comment · Reviewer_zkya · 2022-08-18
> > **Reply**
> >
> > Thanks for addressing my concerns and questions.  After this, I reconsider my score and appreciate the novelty and importance of the benchmark introduced by this paper.

---

> > > ### Author Response · Authors · 2022-08-18
> > > **Reply**
> > >
> > > Thank you very much for your positive response to our rebuttal, and we are really happy that you are satisfied with it now.
> > > For the camera-ready version, we would like to have an additional effort to polish the paper following all the reviewer's encouraging advice, in particular, the ones we cannot fully address because of the time limitation.

---

### Official Review · Reviewer_3WKQ · 2022-07-26
**An extensive benchmark collection for surrogate modeling of physics problems**

**Rating:** 7
**Confidence:** 3

**Strengths:**

- Very timely work: Scientific machine learning is a quickly growing area with little standardized benchmarks. Thus, works like this one are urgently needed to assess new methods in this area.
- The benchmarks cover a wide range of physics (compressible and incompressible flows, waves, 1D-3D problems) that are relevant for a wide range of science and engineering problems.
- The authors provide benchmarks that are otherwise time-consuming to set up because they require knowledge about the underlying physics (boundary conditions etc), the numerical methods to use, and familiarity with software that implements the numerical methods. Thus, the provided benchmarks can save lots of time.
- The authors emphasize error metrics that are more specific to science and engineering problems than standard metrics considered in machine learning (for example, cMSE and bMSE).
- The data sets and repository are well organized and documented.

**Weaknesses:**

- The authors consider a very specific problem of scientific machine learning (see Section 3.1), namely surrogate modeling. Scientific machine learning also investigates techniques for numerically solving PDEs without having training data available from ground-truth solvers, among many other things. I therefore don't think that the title "... for Scientific Machine Learning" is appropriate and instead should be changed to something that includes surrogate modeling rather than scientific machine learning. In that respect, I also don't agree with the statement "The objective of Scientific ML is to find some ML-based surrogate" on page 3, because scientific machine learning is much broader than surrogate modeling.

- Because the focus is on surrogate modeling, there should be at lease one traditional, widely used baseline such as non-intrusive POD models (for example: Guo and Hesthaven, Data-driven reduced order modeling for time-dependent problems, https://doi.org/10.1016/j.cma.2018.10.029). Otherwise the benchmarks don't help to assess if the machine learning methods can provide improved results compared to traditional surrogate modeling techniques.

- The authors compare methods based on accuracy. However, one major purpose of surrogate modeling is to achieve a low runtime of evaluating a model but no runtime results (training, evaluation runtimes) are provided. Having runtime results would help to assess the speedup that can be obtained with various surrogate models.

**Additional Feedback:**

See comments under Weaknesses and Clarity.

**Clarity:**

The paper is well written. Some minor changes could further improve the clarity:
- The function v defined at the beginning of Section 3.1 has no range.
- In Section 3.1, the notation of v(., ., \theta) changes to v_{\theta}() without explanation
- The author should consider introducing a time step size delta t at the beginning of Section 3.1 instead of using t + 1.

**Correctness:**

The supplemental material is extensive and provides detailed problem descriptions. The GitHub repository is well structured and all training data are available. The evaluation metrics are appropriate and go beyond classical evaluation techniques and are motivated by science problems. The selected problems cover a wide range of physics (parabolic and hyperbolic problems, Navier-Stokes)

**Documentation:**

- Scripts to generate new data are provided.
- The authors should consider making the installation of simulation packages such as clawpack optional, because those can require additional packages (fortran compiler) that are uncommon in machine learning and unnecessary if only the pre-computed data are to be used.


**Ethics:**

No concerns.

**Relation To Prior Work:**

The present work looks like a valuable extension of the benchmark suite introduced in [36] by Otness et al. The present work is very similar in style and covers similar physics (fluid dynamics, wave) and similar baselines (U-Net) but provides more benchmark examples and more baseline surrogates than [36], except that no runtimes results are provided as in [36].

**Summary And Contributions:**

The authors introduce a benchmark suite for surrogate modeling, with a focus on deep network models. The benchmark includes problems in one to three dimensions, problems based on Navier-Stokes equations and shallow water equations and several common 1D benchmarks from scientific computing. Simulators for generating new data, pre-trained network models, and pre-computed training data are provided.

---

> ### Author Response · Authors · 2022-08-12
> **Reply to Reviewer #2**
>
> Dear Reviewer #2,
>
> Thank you very much for your kind comment for improving our paper. The following are our responses
>
> >The authors consider a very specific problem of scientific machine learning (see Section 3.1), namely surrogate modeling. Scientific machine learning also investigates techniques for numerically solving PDEs without having training data available from ground-truth solvers, among many other things. I therefore don't think that the title "... for Scientific Machine Learning" is appropriate and instead should be changed to something that includes surrogate modeling rather than scientific machine learning. In that respect, I also don't agree with the statement "The objective of Scientific ML is to find some ML-based surrogate" on page 3, because scientific machine learning is much broader than surrogate modeling.
>
> Thank you very much for your kind comment. As the reviewer pointed out, scientific machine learning covers very broad topics, and creating a surrogate model is one of them. However, we would consider that surrogate modeling is still one of the most important and actively studied topics, so we would like to keep the present paper title. However, instead, we weakened our statement in page 2 (Introduction), and added more explanation about other problems that would benefit from scientific machine learning.
>
> >Because the focus is on surrogate modeling, there should be at least one traditional, widely used baseline such as non-intrusive POD models (for example: Guo and Hesthaven, Data-driven reduced order modeling for time-dependent problems, https://doi.org/10.1016/j.cma.2018.10.029). Otherwise the benchmarks don't help to assess if the machine learning methods can provide improved results compared to traditional surrogate modeling techniques.
>
> Thank you very much for bringing our attention to this line of research. Indeed, a comparison to traditional surrogate modeling techniques would present a nice addition. However, we cannot do all baseline comparisons. First, obtaining trustworthy POD benchmark results for all of our datasets exceeds what we can do within the rebuttal period. Second, our aim is not to advertise a certain ML method and to prove its superiority, but to provide a representative and expressive set of benchmark problems. Comparing different ML methods and showing their reliability is then in a later step what the benchmark was designed for. So far we concentrated on what we considered a representative selection of state-of-the-art SciML methods from a Machine Learning perspective. Please see further our general reply. We hope this reply is still satisfactory.
>
> >The authors compare methods based on accuracy. However, one major purpose of surrogate modeling is to achieve a low runtime of evaluating a model but no runtime results (training, evaluation runtimes) are provided. Having runtime results would help to assess the speedup that can be obtained with various surrogate models.
>
> Thank you very much for your fruitful comment! We have added a new section discussing inference time comparison between machine learning models and simulation methods in Subsection 4.4 and Appendix F, G. Please also refer to our “General Reply” to this issue.
>
>
> >The authors should consider making the installation of simulation packages such as clawpack optional, because those can require additional packages (fortran compiler) that are uncommon in machine learning and unnecessary if only the pre-computed data are to be used.
>
> Thanks for pointing this out. We have now separated the dependencies into multiple requirements.txt files for the different use cases. One is provided for training from pre-computed data while the other also includes the additional dependencies for generating new data.
>
>
> >–The function v defined at the beginning of Section 3.1 has no range.
>
> Thank you for your pointing this out! However, the range of v, in general, depends on problems, so we decided to a provide high-level definition of the range of v in the following sentences.
>
> >–In Section 3.1, the notation of v(., ., \theta) changes to v_{\theta}() without explanation
>
> Following your suggestion, we have polished the corresponding parts.
>
> >–The author should consider introducing a time step size delta t at the beginning of Section 3.1 instead of using t + 1.
>
> Thank you; we did consider this. However, as our models all operate in discrete time, using 1 as the time delta is convenient, since without loss of generality, we may always rescale the time axis so that the time delta is in fact one. Further, since our inference takes place in discrete-time, a discrete time index is appropriate.

---

> > ### Comment · Reviewer_3WKQ · 2022-08-18
> > **Reponse**
> >
> > The authors have done a great job in addressing several of my comments. I stay with my rating that this is a good paper that should be accepted.

---

> > > ### Author Response · Authors · 2022-08-18
> > > **Reply**
> > >
> > > Thank you very much for your positive response to our paper! As we described to the other reviewers, we would like to have an additional effort to polish the paper for the camera-ready following all the reviewer's encouraging advice, in particular, the ones we cannot fully address because of the time limitation.

---

### Official Review · Reviewer_aBDp · 2022-07-26
**Benchmarking learning performance on PDEs**

**Rating:** 7
**Confidence:** 3
**Clarity:** The paper is clearly written.

**Strengths:**

* The benchmark considers a large number number of PDEs for flow problems, time dependent and independent, in 1,2,3D. Some of these are typically very complicated and time consuming to implement for someone with or without the background and time, so the breadth of examples presented here is very useful. Furthermore, the interface seems very nice as it directly works with PyTorch. This standardization also means that extensibility should be very good, which is a big benefit for both people who are interested in phrasing problem and those interested in just solving them.
* The benchmark considers performance at various regimes (parameter values which may give rise to new phenomena), which is very important for generalization and getting these ML methods to the level of being black box solvers.
* The benchmark sets forth new comparison metrics which give a better view at physical consistency than traditional ML error metrics. An interesting discussion of how some methods perform on these is also given.


**Weaknesses:**

* The benchmark does not include a comparison of the machine learning approaches with traditional marching schemes, either implicit or explicit. For a task where there exist very strong methods that already solve this problem with convergence guarantees, it is important to compare with traditional baselines to assess true progress in this task. Although generation was done with these methods (hopefully to high enough accuracy to produce valid ground truth), comparisons can be done with larger time steps.
* The benchmark does not compare the time to get a simulation result, which is even more complicated in this setting given the training + eval time required for machine learning.
* The benchmark does not attempt to do prediction in regions outside of training distribution. This can be important for generalization depending on the setting, and in any case a good point of discussion.
* Arguably, non-rectangular domains are pretty important for flow problems, although this was a stated limitation.


**Additional Feedback:**

NA

**Correctness:**

Data generation seems correct, at least given the details in the supplementary info about the finite difference schemes and deference to external solvers. The authors do a good job of deliberating on physical phenomena and difficulties of numerical solutions in various regimes of parameter values. It would be good for the authors to clarify how they ensure that the data generated has sufficient accuracy to be a valid ground truth.


**Documentation:**

Data generation is clearly discussed, persistence seems good (even though you can generate it as you need it) and the dataset seems very accessible given its straightforward and standardized API.

**Relation To Prior Work:**

There is a good discussion of prior work, although this work does not stand as a strict superset of previous efforts to create a benchmark for learning physical simulation. Otness et al 2021, while more limited in the number of PDEs examined, attempts a comparison with traditional simulators through coarsening time steps and trying to match runtime (which is one of the weaknesses here).


**Summary And Contributions:**

The paper presents a benchmark for using machine learning to solve PDEs. The PDEs considered vary from relatively simple to complex. In addition to the data, the authors provide a framework for data generation that can be extended with different ranges of parameters and different equations.

---

> ### Author Response · Authors · 2022-08-12
> **Reply to Reviewer #1**
>
> Dear Reviewer #1,
>
> Thank you very much for your kind comment for improving our paper. The following are our responses.
>
> >The benchmark does not include a comparison of the machine learning approaches with traditional marching schemes, either implicit or explicit. For a task where there exist very strong methods that already solve this problem with convergence guarantees, it is important to compare with traditional baselines to assess true progress in this task. Although generation was done with these methods (hopefully to high enough accuracy to produce valid ground truth), comparisons can be done with larger time steps.
>
> Thank you for the suggestion. We added Appendix H, where we performed a small experiment by generating a higher resolution dataset for the 2D diffusion-reaction problem and downsampled the data to train the ML models. Then, we also generated a lower resolution dataset to be compared to the downsampled data, that we assumed as the ground truth. This way, we can compare the prediction error of the lower resolution data with the downsampled high resolution data, to evaluate the improvement obtained by the ML models predictions. Please also see the problem selection section in our general response to all reviewers.
>
> >The benchmark does not compare the time to get a simulation result, which is even more complicated in this setting given the training + eval time required for machine learning.
>
> Thank you very much for your fruitful comment! We have added a new section (Section 4.4 and Appendix F,G) discussing inference time comparison between machine learning models and the traditional simulation methods we used (Please also check our “General Remarks”). Although we did not use the recent state-of-the-art numerical methods, we believe our simulation methods can provide a representative baseline in comparison to ML methods. Comparing with the state-of-the-art simulation methods would be in our future work.
>
> >The benchmark does not attempt to do prediction in regions outside of training distribution. This can be important for generalization depending on the setting, and in any case a good point of discussion.
>
> Thank you for your kind suggestion! Unfortunately, neither FNO nor Unet can accept PDE parameters externally, so they do not have generalization power in principle (Note that initial and boundary conditions do not include any PDEs’ information, so ML models cannot understand which PDE or parameters should be used without additional information, e.g., parameter embedding). Although there are several ML models implementing parameter embedding, e.g., PINO, HyperPINN, and message-passing networks for simulation, we feel the research is still immature to propose a baseline score. Hence, we would like to avoid treating the generalization dependence in our paper.
>
> >Arguably, non-rectangular domains are pretty important for flow problems, although this was a stated limitation.
>
> Thank you very much for your encouraging suggestion! Unfortunately many ML models for solving PDEs are CNN-based and therefore can only treat rectangular meshes, so we decided only to include rectangular domains for the moment. Our purpose is to provide benchmark datasets that are widely applicable to as many models as possible. However, again we would like to agree that including non-rectangular domains is the necessary next step for the community, and would like to include them in the future.

---

> > ### Comment · Reviewer_aBDp · 2022-08-24
> > **Response**
> >
> > Thanks for addressing my comments, it seems that they were resolved with more experiments and results. I keep the rating that this is a good paper that should be accepted.

---

> > > ### Author Response · Authors · 2022-08-24
> > > **Response**
> > >
> > > Thank you very much for your positive response to our paper!
> > > Thanks to your encouraging comments, our paper has been significantly improved.
> > >
> > > As we described to the other reviewers, we would like to have an additional effort to polish the paper for the camera-ready following all the reviewer's encouraging advice, in particular, the ones we cannot fully address because of the time limitation.

---

### Author Response · Authors · 2022-08-12
**General Reply to All the Reviewers**

Dear all the reviewers,

Thank you very much for your very kind and encouraging suggestions! We have modified our paper following your advice, which has also been updated in OpenReview. The modified part is colored in red to save your time to find those locations. In this general response, we summarize our answers to common concerns shared by most (if not all) of the reviewers.

## Runtime comparison

First of all, we found that all the reviewers are interested in the comparison between the classical simulation and ML methods. Following those advice, we added the following new sections: Section 4.4: “Inference Time Comparison”, Appendix F: “Detailed Runtime Comparison”, and Appendix G: “Resolution Sensitivity of Inference Time”, where we provided the information on the measured inference time and sensitivities of sub-groups of the PDEs.

In the table below, we provide a part of the comparison between the numerical PDE solvers that we use to generate a single data sample and the FNO which in our experiments is the most accurate ML model. We also list the ML model total training time as additional information. All the time values presented here are defined in seconds. The highest computational demand originates from the ML training time. However, once the ML model was trained, we obtained improvement in the computation time by multiple orders of magnitude. For a complete overview, please see Appendix F and G.

\\begin{array}{l|r|r|r|r}
\hline
  \text{PDE} & \text{Spatial Resolution} & \text{Simulation time per instance [s]}  & \text{ML inference time per instance[s]} & \text{ML training time over all instances [s]}  \\\\
\hline
  \text{1D diffusion-sorption} & 1\,024^1 & 59.83 & 0.32 & 48\,760 \\\\
  \text{2D CFD} (\eta=\zeta=0.1)& 512^2  & 582.61  & 0.14 & 107\,567  \\\\
  \text{3D CFD} (\text{inviscid})& 64^3 & 60.06 & 0.14 & 12\,387  \\\\
\hline
\\end{array}

## Problem selection

Another common concern that we want to address is regarding the problem selection, or the completeness of our benchmark. We have not added many additional experiments beyond this, despite various comments suggesting specific new ones. Each of the reviewers has a favoured set of experimental tasks that are “missing” from the paper, whether that is a comparison with a particular neural model, a particular baseline method (POD, etc), a particular trade-off of training time, simulation time or resolution, or target inference task. Many of these tasks we agree would be informative to perform, and individually each of these suggestions is valid. However, collectively we cannot accommodate all of these requests and their combinations and permutations. The number of combinations of each of these tasks would rapidly lead to a combinatorial explosion of total experiments to be performed, and in any case, would be beyond the scope of this paper. Our goal, in preparing a benchmark, is not to then use the benchmark to solve all possible combinations of inference tasks on all known experiments, but rather to ease the task for subsequent researchers to benchmark their favoured methods. We need to draw a line around the experiments we do perform, and this line must be an arbitrary one, and cannot please everyone; there will always be one more thing missing. Thus we have drawn that line around the experiments that we feel best demonstrate the usefulness of this dataset for performing general benchmarking experiments without pretending that we can be exhaustive in the benchmarking experiments that we ourselves run. As a rule of thumb, we have favoured models that are popular and tasks that we feel have been under-researched, and that is suggestive of interesting new research directions. Part of our goal here is to invite other researchers to fill in the gaps for themselves by leveraging our ready-to-run models to perform their own favoured tasks. We have modified the introduction to clarify our position on this trade-off.

Finally, we would like to emphasize that our repo and dataset have already been installed and tried by many researchers. We would like to consider this as a sign that our “PDEBench” is actually fruitful for many researchers in this community.

---

### Meta-Review · Area_Chair_BbAH · 2022-09-13

**Recommendation:** Accept
**Confidence:** 4

**Metareview:**

The paper proposes a new benchmark consisting of a variety of PDE problems. Almost all the reviewers appreciated the extensiveness of the problems included: time-(in)dependent, 1-3D etc. There is code for generating data, training baseline models, recording variety of metrics -- all the qualities that make for a complete and useful benchmark. The reviewers wanted to see some evaluations comparing ML models with classic approaches, which the authors have duly incorporated during the discussion period. Considering all these aspects, coupled with the significant opportunity for Scientific ML to solve challenging real-world problems in the near and long term, I am optimistic this benchmark will aid ML and PDE communities immensely and lower the bar for entry for researchers and practitioners alike.

---

### Decision · Program_Chairs · 2022-09-16

Accept